# Platelet, Fibrinolytic and Other Coagulation Abnormalities in Newly-Diagnosed Patients with Chronic Thromboembolic Pulmonary Hypertension

**DOI:** 10.3390/diagnostics12051238

**Published:** 2022-05-16

**Authors:** Eleni Vrigkou, Argirios Tsantes, Dimitrios Konstantonis, Evdoxia Rapti, Eirini Maratou, Athanasios Pappas, Panagiotis Halvatsiotis, Iraklis Tsangaris

**Affiliations:** 1Second Department of Critical Care Medicine, Attikon University Hospital, School of Medicine, National and Kapodistrian University of Athens, 12462 Athens, Greece; elenivrigkou@gmail.com (E.V.); d.konstantonis@gmail.com (D.K.); pappasath3@gmail.com (A.P.); 2Laboratory of Hematology and Blood Bank Unit, Attikon University Hospital, School of Medicine, National and Kapodistrian University of Athens, 12462 Athens, Greece; atsantes@med.uoa.gr (A.T.); evirapti7@gmail.com (E.R.); 3Laboratory of Clinical Biochemistry, Attikon University Hospital, School of Medicine, National and Kapodistrian University of Athens, 12462 Athens, Greece; maratueirini@gmail.com; 4Second Department of Internal Medicine, Attikon University Hospital, School of Medicine, National and Kapodistrian University of Athens, 12462 Athens, Greece; pahalv@gmail.com

**Keywords:** chronic thromboembolic pulmonary hypertension, fibrinolysis, p-selectin, platelets, serotonin, thrombin, thromboxane A2, von Willebrand

## Abstract

The pathophysiological background of chronic thromboembolic pulmonary hypertension (CTEPH) has not been fully elucidated. Evidence suggests that abnormal platelet function and ineffective fibrinolysis may play a key role in the development of the disease. The purpose of this study was to evaluate platelet and coagulation function in CTEPH, using non-conventional global coagulation assays, and platelet activation and endothelial dysfunction laboratory markers. A total of 40 newly-diagnosed CTEPH patients were studied, along with 35 healthy controls. Blood samples from CTEPH patients were taken directly from the pulmonary artery. All subjects were assessed with platelet function analyzer-100, light transmission aggregometry, thromboelastometry, endogenous thrombin potential. von Willebrand antigen and activity, p-selectin, thromboxane A2 and serotonin levels were also assessed. The results showed that CTEPH patients present diminished platelet aggregation, presence of disaggregation, decreased rate of fibrinolysis, defective thrombin generation and increased levels of thromboxane A2, p-selectin, von Willebrand antigen and activity. Serotonin levels did not present any differences between the two groups. The results of this study suggest that CTEPH patients present platelet function, fibrinolytic, thrombin generation and other clot formation abnormalities. Well-designed clinical studies are needed to further evaluate the complex hemostatic abnormalities in the CTEPH setting and assess their potential clinical applications.

## 1. Introduction

Chronic thromboembolic pulmonary hypertension (CTEPH) is a progressive, potentially life-threatening pulmonary vascular disease, categorized within group 4 of the current classification of pulmonary hypertension [1]. It is caused by pulmonary artery and arteriole thromboembolic obstruction due to incomplete-resolved thrombi that result in organized, fibrotic occlusions in the pulmonary vasculature [2]. These obstructions limit the blood flow through the pulmonary vessels, leading to increased pulmonary arterial pressure, pulmonary vascular resistance and right-heart load [3].

CTEPH is a rare, chronic disorder, whose global incidence and prevalence rates have not been yet clearly assessed [4]. It has been estimated, nonetheless, that the annual incidence of CTEPH is approximately 4 cases/100,000 population [4]. CTEPH is viewed as a long-term complication of pulmonary embolism (PE), affecting approximately 3% of PE survivors [5,6]. The disorder can be diagnosed when the clots in the pulmonary vasculature have not been resolved after more than three months of effective anticoagulant treatment [1]. CTEPH, however, can occur months or years after the acute PE, without the development of a new thromboembolic event in the meantime [4].

CTEPH’s pathophysiological background is considered to be much more complicated than a poorly resolved PE [7]. This hypothesis is based mainly on observations, such as the facts that the majority of PE survivors don’t develop CTEPH, that a history of acute PE or deep vein thrombosis is lacking in around 30% of proven CTEPH patients and that CTEPH could persist even after the performance of pulmonary thromboendarterectomy [7,8].

Despite research advances, several aspects of the pathogenetic mechanisms of the disease remain elusive to date [9]. Initially, it was thought that the unresolved thrombi were the cause of CTEPH [10]. It is now considered that endothelial dysfunction, vascular remodeling and vasoconstriction participate in the etiopathogenesis of the disease [3]. More specifically, evidence suggest that there is secondary microvasculopathy in the pulmonary vasculature of CTEPH patients, in the form of obstructive (but non-thrombotic) remodeling of the small pulmonary arteries, capillaries and venules [10]. These lesions appear to be similar to lesions found in patients suffering from pulmonary arterial hypertension (e.g., plexiform lesions, intimal fibrosis and proliferation) [10]. Literature data also support that abnormal platelet function and ineffective fibrinolysis may play a key role in the development of the disorder [9,11]. Nevertheless, original research data portraying platelet and coagulation abnormalities related to CTEPH remain limited.

In a previous study conducted by our group [12], we assessed platelet and coagulation processes in newly-diagnosed patients with pulmonary arterial hypertension using non-conventional coagulation assays, that portrayed platelet dysfunction and thrombin generation deficits in the patient population. Since there could be common mechanisms that contribute to the development of both pulmonary arterial hypertension and CTEPH that could affect hemostasis (e.g., endothelial dysfunction and vascular remodeling) [10], in this study we assessed platelet and coagulation processes in the CTEPH population. For this purpose, we collected blood directly from the pulmonary artery of newly-diagnosed CTEPH patients, before the administration of PH-specific medications. Blood samples were processed using the following assays: platelet function analyzer-100 (PFA-100), light transmission aggregometry (LTA), rotational thromboelastometry (ROTEM) and endogenous thrombin potential (ETP). Platelet activation and endothelial dysfunction laboratory markers (serotonin, thromboxane A2 (TXA2), soluble p-selectin, and von Willebrand antigen (VWF:Ag) and activity(VWF:Ac)) were also assessed from the same samples.

## 2. Materials and Methods

### 2.1. Participants

40 consecutive CTEPH patients were recruited from the Pulmonary Hypertension Unit of Attiko University Hospital of Athens at the time of diagnosis. CTEPH patients had received more than three months of therapeutic anticoagulant therapy. They were currently on bridging therapy with low molecular weight heparin due to planned right-heart catheterization for the initial diagnosis of their disease. This therapy was stopped at least 24 h prior to the performance of the catheterization. CTEPH diagnosis was performed according to European Society of Cardiology and European Respiratory Society guidelines [1]. The control group consisted of 35 healthy volunteers and blood donors, age and sex-comparable to the patient population. None of the healthy participants received anticoagulant, antiplatelet or any other drug therapy. The exclusion criteria of this research were: hepatic and renal insufficiency, malignancy, active infections, and use of other medication known to affect platelet function and coagulation. The study was conducted in accordance with the amended Declaration of Helsinki.

### 2.2. Sample Collection Method

Blood samples were obtained from the patients’ pulmonary artery during a right-heart catheterization performed for the diagnosis of CTEPH. More specifically, the collection was made from the distal lumen of the pulmonary catheter with very slow aspiration. Blood samples from the healthy participants were obtained from the antecubital vein. The blood specimens were placed in blood collection tubes containing 3.8% trisodium citrate.

### 2.3. Coagulation Assays

All blood specimens were processed by PFA-100, LTA, ROTEM and ETP. Markers of platelet activation and endothelial dysfunction (serotonin, TXA2, VWF:Ac, VWF:Ag and soluble p-selectin) were also assessed from the same samples. PFA-100, LTA and ROTEM assays were performed within three hours from the collection of the specimens. The ETP assay and the tests assessing platelet activation and endothelial dysfunction laboratory markers were performed from plasma obtained after the whole blood samples were centrifuged. The removed plasma was snap frozen and stored in laboratory fridges at the temperature of −20 °C (with round-the-clock monitoring of the freezer temperatures). These tests were performed at a later time.

#### 2.3.1. PFA-100

Primary hemostasis was assessed by PFA-100 (Dade Behring, Marburg, Germany). The purpose of this assay is to activate platelets using shear stress and agonists [13]. For this test, we used a 0.8 mL sample of whole blood. As an agonist, epinephrine (EPI) was utilized. The blood sample was placed in a cartridge containing EPI. During this test, the activation of the platelets results in the formation of platelet aggregates, that finally close an opening located in the cartridge. The PFA-100 analyzer calculates the time needed for this opening to be closed (closure time (CEPI CT)).

#### 2.3.2. LTA

Platelet function was assessed by an LTA assay (Biodata-PAP-4 instrument, Bio/Data Corp., Horsham, PA, USA). In this test, light is being transmitted through a sample of the patient’s plasma. The analyzer evaluates changes in that transmission during the formation of platelet aggregates [14]. More specifically, platelets are activated by an agonist and then begin to aggregate. As the platelets aggregate, the transmission of light through the blood sample increases. The results of the LTA testing are presented as a trace and the measured parameter is maximal aggregation (%). In Figure 1 we present an example of an aggregation trace as it is produced by the aggregometer. To set the baseline (0% aggregation), platelet rich plasma is used (centrifugation of a whole blood sample for 10 min at 200× *g*). To establish the 100% aggregation, platelet poor plasma is used (re-centrifugation of the sample for 15 min at 2000× *g*).

Two tests were performed in the LTA analyzer. For the first test we used 0.05 mL of ADP 2.0 × 10^−5^ M as an agonist and for the second 0.05 mL of EPI 1.0 × 10^−4^ M. The assay was left to run for 10 min. In the cases where disaggregation was present, the disaggregation value was calculated by the difference between the maximal and final aggregation values. Disaggregation is indicative of the stability of platelet aggregates [15]. An example of a disaggregation trace is presented in Figure 2.

#### 2.3.3. Rotem

The Rotem assay (Tem Innovations GmbH, Munich, Germany) is a point-of-care test designed to assess the viscoelastic properties of clot formation and dissolution [16]. It essentially provides a complete profile of clot development (including kinetics of clotting, clot strength, fibrinolysis, and assessment of interactions between platelets, coagulation factors and inhibitors) [16]. The results of this assay are depicted as a trace. An example of the Rotem trace is presented in Figure 3.

Rotem’s major measured parameters are presented and briefly interpreted in Table 1.

For this study, we performed a non-activated TEM assay, using 20 μL of CaCl_2_ 0.2 mol/L as a reagent.

#### 2.3.4. ETP

ETP (INNOVANCE^®^, Siemens Healthcare Diagnostics, Marburg, Germany) assesses plasma’s capacity to generate thrombin (total amount and time course) [19]. The measured parameters of this assay are:The time needed until the start of thrombin generation (lag time/tlag),The total amount of thrombin generated (area under the curve/AUC),The maximum thrombin concentration (Cmax), andThe time needed to reach maximum thrombin generation (tmax).

In this assay, phospholipids, calcium ions, and human recombinant tissue factor are utilized to activate coagulation and, thus the generation of thrombin. The process is recorded with the aid of a slow reacting chromogenic substrate (at 405 nm) and a fibrin aggregation inhibitor.

#### 2.3.5. Markers of Platelet Activation and Endothelial Dysfunction

*TXA2:* TXA2 is produced by activated platelets and is a well-known marker of platelet activation [20]. TXA2 levels were assessed using an enzyme-linked immunosorbent assay (American Research Products, Inc., Waltham, MA, USA).

*p-selectin:* p-selectin functions as an adhesion molecule, and is stored in the granules of platelets and in the Weibel–Palade bodies of an endothelial cell [21]. It is frequently used in research as a marker of both platelet activation and endothelial dysfunction [21]. The levels of p-selectin were evaluated by an enzyme-linked immunosorbent assay (Young in Frontier Co, Ltd., Seoul, Korea).

*Serotonin:* Serotonin is produced by activated platelets [22]. In the coagulation setting, serotonin promotes vasoconstriction and platelet aggregation [22]. For the purposes of serotonin measurements, the enzyme-linked immunosorbent assay kit used was manufactured by Enzo Life Sciences (New York, NY, USA).

*VWF:Ag and VWF:Ac*: von Willebrand factor, a glycoprotein synthesized by endothelial cells, facilitates platelet adhesion to the injured vascular subendothelium [23]. It is widely used as a marker of endothelial dysfunction in vascular disorders [23]. An automated latex enhanced immunoassay (HaemosIL™ assay, Instrumentation Laboratory Company, Lexington, KY, USA) performed on IL(Instrumentation Laboratory) coagulation systems was used to evaluate VWF:Ag and VWF:Ac. In this assay, latices covered with an anti-VWF monoclonal antibody react with the vWF, creating particle aggregates that can be detected by turbidimetry. The assay’s results are provided as percentage of normality.

#### 2.3.6. Conventional Coagulation Tests

A Sysmex XE-2100 analyzer (Roche, Chicago, IL, USA) was used to determine complete blood counts. Fibrinogen levels were measured by a modified Clauss method (Siemens Healthcare Diagnostics, Marburg, Germany). A particle-enhanced immunoturbidimetric method (Innovance, Siemens Healthcare Diagnostics, Marburg, Germany) was utilized for the detection of the D-dimers. Lastly, aPTT, PT and INR were processed by a BCS^®^ XP System Hemostasis analyzer (Siemens Healthcare Diagnostics, Marburg, Germany).

#### 2.3.7. Statistical Analyses

Statistical analyses were performed in Stata software (Stata Corp., College Station, Texas, TX, USA) using the two sample Wilcoxon rank-sum test. All tests were two-sided. The results were judged as statistically significant when the probability level was <0.05.

## 3. Results

In Table 2 we report the demographics and standard hematological test for all of the participants in the study, along with hemodynamic measurements and clinical findings of the CTEPH population.

The results of platelet, coagulation and endothelial parameters for all study participants are reported in Table 3 and Table 4.

*PFA-100:* The PFA-100 test, which assesses primary hemostasis, did not reveal any statistically significant results in the CEPI CT parameter between the two groups.

*LTA assay:* CTEPH patients presented defective platelet aggregation in the LTA testing when both EPI (*p* < 0.001) and ADP (*p* < 0.001) were used as agonists. The results are reported as Box and Whisker plots in Figure 4. Additionally, 60% of the patient population showed disaggregation in the LTA ADP assay. Disaggregation is indicative of unstable platelet aggregate formation and is not normally seen in healthy individuals [15].

*ROTEM*: The ROTEM testing revealed prolonged initiation of the clotting process (CT: *p* = 0.02) and clot propagation (CFT: *p* < 0.001) in CTEPH patients compared to controls. The a angle, which reflects the speed of clot formation, was found to be defective in the patient group (*p* = 0.01). The rate of fibrinolysis was decreased in the patient population in relation to healthy controls (Li60: *p* = 0.04). The MCF parameter did not present any significant differences between the two groups. The statistically significant results of this assay are presented as Box and Whisker plots in Figure 5.

*ETP assay*: The ETP assay showed that thrombin generation is defective in the patient group in relation to healthy individuals; Both the maximum thrombin concentration (Cmax: *p* = 0.03) and the total amount of thrombin generated (AUC: *p* < 0.001) where found to be diminished. The respective Box and Whiskers plots are reported in Figure 6. No other ETP parameters presented any differences between the two groups.

*Platelet activation and endothelial dysfunction markers:* TXA2 and p-selectin levels were increased in CTEPH patients (*p* = 0.006 and *p* = 0.04, respectively), while serotonin levels did not present any differences between the two groups. Lastly, VWF:Ac and VWF:Ag, were found to be increased in the patient group (*p* < 0.001 for both values). The statistically significant results are displayed as Box and Whiskers plots in Figure 7.

## 4. Discussion

The results of this study suggest that platelet function, fibrinolysis, and other aspects of the coagulation process may be defective in CTEPH patients. More specifically, platelet aggregation was shown to be diminished, a significant proportion of the patients presented decreased platelet aggregate stability, the initiation of the clotting process and clot propagation were both prolonged, the rate of fibrinolysis was diminished, and thrombin generation capacity was defective. Regarding platelet activation and endothelial dysfunction markers, TXA2, p-selectin, VWF:Ac and VWF:Ag were found to be increased.

Agonist-induced platelet aggregation in LTA testing was shown to be diminished in the patient population compared to controls. This finding corresponds with other studies in the field showing decreased platelet aggregation after standard agonist concentrations were used [24]. Remková et al. [24] studied platelet aggregation in CTEPH patients using both standard and lower threshold concentrations of ADP and EPI as agonists. They demonstrated that platelet aggregation was diminished in CTEPH patients compared to healthy subjects after standard concentrations of agonists were used (as it was conducted for the present study), but normal when low concentrations of agonists were utilized. The reasons behind this “paradoxal” platelet behavior have not yet been identified [24].

The observed diminished platelet aggregation could be explained by a chronic platelet activation in the CTEPH population [25]. When platelets are chronically activated, there is a prolonged granule release reaction and extended degranulation, that can result in a decreased platelet aggregation ability [24]. There is evidence in the literature supporting that the platelets of CTEPH patients are highly activated; studies have shown that CTEPH patients present increased mean platelet volume, raised percentage of activated GPIIb/IIIa-positive platelets, and elevated GTP-bound levels of RalA [26]. GTP-bound RalA have been shown to control the degranulation process [27].

In the present study markers of platelet activation were also assessed. TXA2 is released by activated platelets and has prothrombotic and vasoactive properties [28]. In the study’s CTEPH population, the levels of TXA2 were found to be elevated. p-selectin was also found to be elevated in the CTEPH group compared to healthy participants. In a study conducted by Yaoita et al., the authors showed that their CTEPH patient group presented elevated p-selectin–positive platelets compared to controls [26].

We additionally assessed serotonin as a marker of platelet activation. No differences were found, however, between the control and the patient groups. A study conducted by Ulrich et al., that evaluated serotonin levels in a mixed population of CTEPH and pulmonary arterial hypertension patients, showed no differences in serotonin values between the patient and control groups [29]. The authors hypothesized that this finding could be explained by an increased consumption or degradation of serotonin in the lungs of pulmonary hypertension patients [29].

The ROTEM assay revealed several abnormal parameters. The CT value, which depends on thrombin formation, was found to be prolonged in CTEPH patients. This outcome corresponds with the outcomes of the ETP assay that showed that the total amount of free thrombin and Cmax were diminished. The CFT parameter depends on fibrinogen levels and platelet-based clot stability [17]. Since fibrinogen values in the CTEPH group were not decreased, the prolonged CFT parameter could reflect platelet- based clot instability. This finding is supported by the disaggregation observed in the LTA ADP tracing in CTEPH patients, which indicates unstable platelet aggregate formation. Additionally, the defective a-angle values, which represent the kinetics of clotting, also reflect diminished clot stability. On the contrary, the results of PFA-100 testing did not show any differences in primary hemostasis between patients and controls. This could be explained by the fact that PFA-100 can produce normal results in platelet dysfunction related to storage pool deficiencies [13].

The strength of the fibrin-based clot, as it is depicted by the MCF parameter, appears to be analogous in both CTEPH patients and healthy controls. This finding suggests that despite the above-mentioned platelet function abnormalities, the stability of the final, fibrin-based clot is within normal ranges. The increased Li60 parameter is indicative of a decreased rate of fibrinolysis. Several studies have associated thrombin generation abnormalities with the production of abnormally structured clots and resistance to fibrinolysis (through changes in fiber thickness and density of the fibrin clot) [30]. In the present study, CTEPH patients presented defective thrombin generation (assessed by the ETP) and decreased fibrinolysis (evaluated by the Li60 parameter). Data in the international literature suggest that fibrin derived from CTEPH patients may be resistant to lysis and that could be an important part of the pathophysiology of the disease [31].

The fibrinogen concentrations of the patient group were increased compared to controls. Other studies have also found elevated fibrinogen levels in the CTEPH population and they have linked them to disease severity [32]. Hennigs et al. also suggested that fibrinogen plasma concentration may be an independent marker of hemodynamic impairment in CTEPH [32]. On the contrary, d-dimer levels showed no statistically significant results between the two groups. Other research studies have also found d-dimer levels in CTEPH patients comparable to healthy subjects [24,26]. D-dimers are degraded products of fibrin fibers. CTEPH is a chronic, “fibrotic” disorder, where clots eventually become integrated into the vascular wall, leaving less fibrin products (such as d-dimers) available to be detected [33]. Lastly, VWF: Ag and VWF: Ac were assessed as markers of endothelial dysfunction in this study and they were found to be elevated in CTEPH patients compared to controls. This outcome comes in accordance with several other studies in the field showing the same results [24,26,32].

In our previous study where we studied platelet and coagulation processes in pulmonary arterial hypertension [12], we showed that the patient population also presented diminished platelet aggregation, prolonged initiation of the clotting process and thrombus formation, and diminished total amounts of free thrombin and maximum thrombin concentrations. Since the pathogenetic processes of both pulmonary arterial hypertension and CTEPH have not been clarified to date and the role that coagulation deficits play in the development and progression of pulmonary hypertension have not been elucidated [4,10], it could be difficult to speculate if those findings are characteristics of pulmonary hypertension or an epiphenomenon [34].

To our knowledge, this is the first study that evaluates globally the coagulation process (platelet function, fibrinolysis, thrombin generation, viscoelastic properties of clot formation, markers of platelet activation and endothelial dysfunction) of newly-diagnosed CTEPH patients, before the administration of PH-specific medication. One of this study’s main limitations is the small number of included patients. Nevertheless, our sample size is analogous with other similar studies in the field [24,26].

## 5. Conclusions

The results of this study suggest that CTEPH patients present platelet function, fibrinolytic, thrombin generation and other clot formation abnormalities. Even though the research on the field of the pathobiological background of the disorder is rapidly evolving, the complex effects that coagulation may have on the progression and development of the disorder have not been yet understood. Well-designed clinical studies are needed to further evaluate hemostatic abnormalities in the CTEPH setting and assess their potential clinical applications.

## Figures and Tables

**Figure 1 diagnostics-12-01238-f001:**
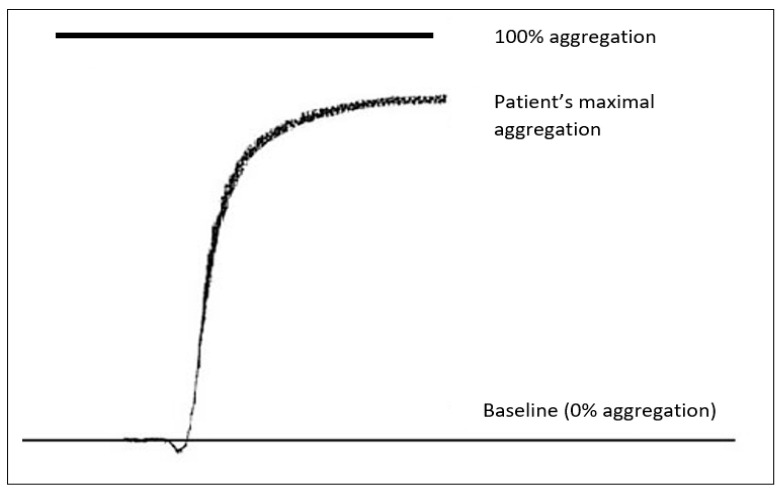
Graphical illustration of a normal LTA aggregation trace.

**Figure 2 diagnostics-12-01238-f002:**
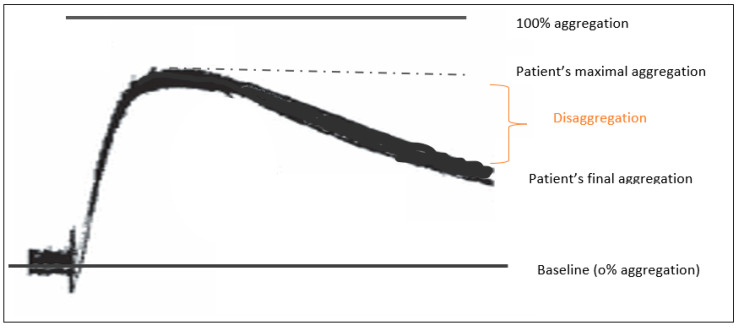
Graphical illustration of an LTA trace showing disaggregation.

**Figure 3 diagnostics-12-01238-f003:**
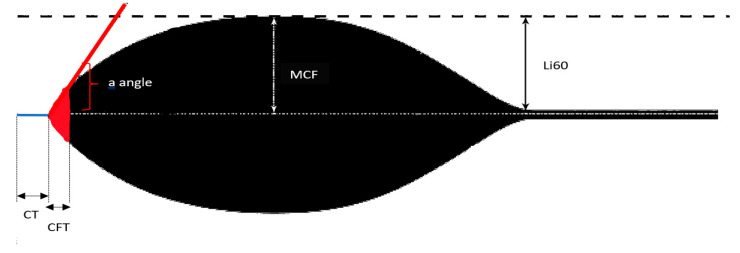
Graphical illustration of a normal Rotem trace.

**Figure 4 diagnostics-12-01238-f004:**
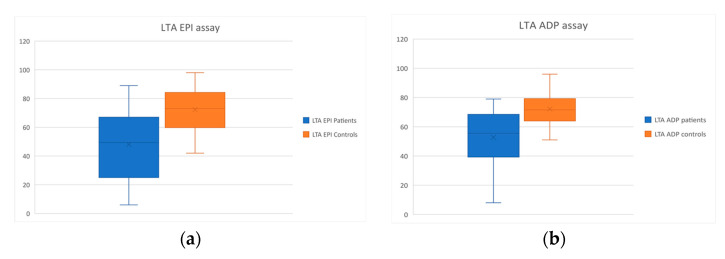
Box and Whisker plots of the results of the LTA assays of the patient and control groups. (**a**) LTA assay performed using EPI as agonist; (**b**) LTA assay using ADP as agonist.

**Figure 5 diagnostics-12-01238-f005:**
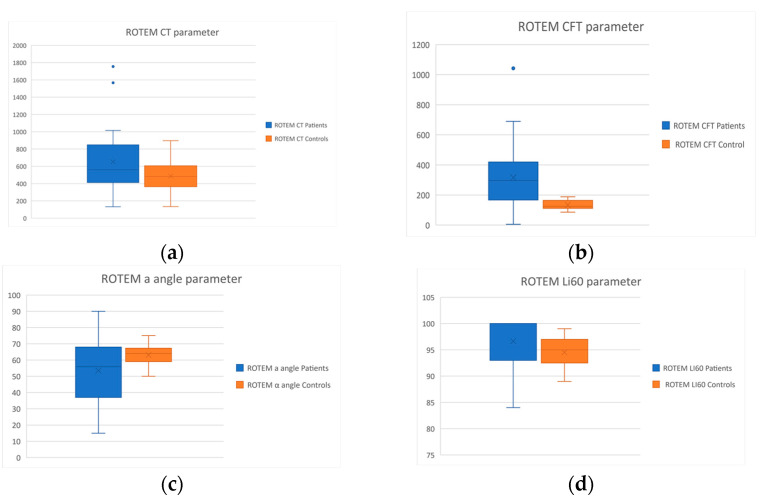
Box and Whisker plots of the results of the ROTEM assay of the patient and control groups. (**a**) depiction of the CT parameter; (**b**) depiction of the CFT parameter; (**c**) depiction of the a angle parameter; (**d**) depiction of the Li60 parameter.

**Figure 6 diagnostics-12-01238-f006:**
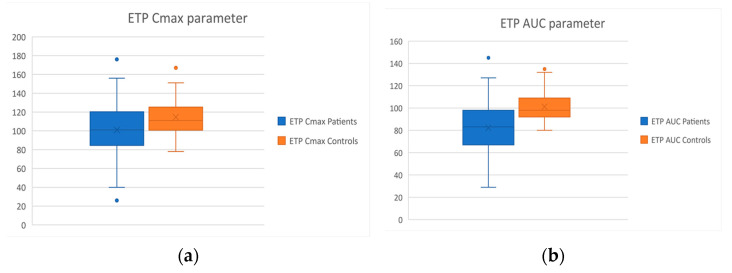
Box and Whisker plots of the results of the ETP assay of the patient and control groups. (**a**) depiction of the Cmax parameter; (**b**) depiction of the AUC parameter.

**Figure 7 diagnostics-12-01238-f007:**
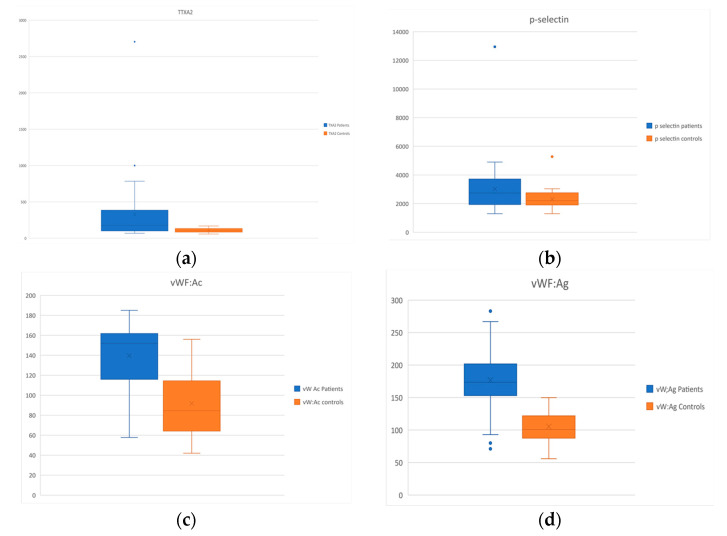
Box and Whisker plots of the results of the assays assessing markers of platelet activation and endothelial dysfunction. (**a**) TXA2; (**b**) p-selectin; (**c**) vWF:AC; (**d**) vWF:Ag.

**Table 1 diagnostics-12-01238-t001:** Rotem’s major parameters and their interpretation [17,18].

Rotem Parameters	Interpretation
Clotting time (CT)	Time needed for clotting to start; it assesses thrombin formation and the initition of clot polymerization
Clot formation time (CFT)	Time needed to reach a specific clot strength (20 mm); It assesses fibrin polymerization and clot stability by platelets and factor XIII
a angle	Estimates the speed of thrombus formation (clotting kinetics)
Maximum clot firmness (MCF)	Represents the ultimate strength of the thrombus; It assesses final clot stability by the polymerized fibrin, platelets and factor XIII
Lysis index at 60 min (Li60)	Represents the stability of the thrombus (compared to MCF) 60 min after CT; It assesses the rate of fibrinolysis

**Table 2 diagnostics-12-01238-t002:** Demographics and standard hematological tests of study participants. Hemodynamic measurements and clinical findings of the CTEPH population.

	Healthy Controls (*n* = 35)	CTEPH Patients (*n* = 40)
**Age (years)**	59.9 ± 11.9	60.1 ± 15.9
**Female (%)**	21/35 (60%)	26/40 (65%)
**Caucasian Race (%)**	35/35 (100%)	40/40 (100%)
**PLTs (10^3^/μL)**	250 ± 46	268 ± 54
**WBC (/μL)**	6750 ± 1010	6268 ± 1640
**Hb (g/dL)**	14.3 ± 1.4	13.2 ± 2.0
**INR**	1 ± 0.08	1.1 ± 0.2
**aPTT (sec)**	30.4 ± 3.5	32 ± 3.9
**Creatinine (mg/dL)**	0.8 ± 0.1	1 ± 0.2
**AST (U/L)**	21.2 ± 13.4	17 ± 6.7
**ALT (U/L)**	20.7 ± 11.5	18 ± 8.0
**mPAP (mm Hg)**	-	40.7 ± 16.5
**PVR (Wood units)**	-	9.4 ± 4.7
**CI (L/min/m^2^)**	-	2.2 ± 0.5
**NT-proBNP (pg/mL)**	-	1148 ± 1734
**6MWT (m)**	-	378 ± 102

**Table 3 diagnostics-12-01238-t003:** Platelet and coagulation parameters in CTEPH patients and healthy controls.

	Healthy Controls	CTEPH Patients	Significance
**CEPI CT (sec)**	131.6 ± 17.9	142.6 ± 58.7	*p* = 0.43
**LTA Epi (%)**	72.4 ± 15.1	48.1 ± 22.6	*p* < 0.001
**LTA ADP (%)**	72.1 ± 10	52.7 ± 18.3	*p* < 0.001
**Patients (%) with disaggregation**	0/35 (0%)	24/40 (60%)	*p* < 0.001
**CT (sec)**	486.7 ± 168.9	651.9 ± 366.9	*p* = 0.02
**CFT (sec)**	133.5 ± 29.8	316.4 ± 203.1	*p* < 0.001
**a (^o^)**	63.1 ± 6.3	53.5 ± 20.1	*p* = 0.01
**MCF (mm)**	58.1 ± 5.7	56.4 ± 16.6	*p* = 0.57
**Li60 (%)**	94.5 ± 2.7	96.6 ± 5.1	*p* = 0.04
**Lag time (sec)**	30 ± 5.2	30.1 ± 5.4	*p* = 0.97
**Tmax (sec)**	83.7 ± 16	82 ± 18.7	*p* = 0.90
**Cmax (%)**	114.7 ± 20.3	101 ± 29.7	*p* = 0.03
**AUC (%)**	114.6 ± 20.3	82.3 ± 26.7	*p* < 0.001

**Table 4 diagnostics-12-01238-t004:** Endothelial and coagulation variables in CTEPH patients and healthy controls.

	Healthy Controls	CTEPH Patients	Significance
**Serotonin (ng/mL)**	242.9 ± 96.9	267.9 ± 174.7	*p* = 0.47
**Thromboxane A2 (pg/mL)**	108.1 ± 31.9	327.8 ± 449.6	*p* = 0.006
**Soluble p-selectin (pg/mL)**	2314.6 ± 686	3019.9 ± 1888.6	*p* = 0.04
**vW:Ac (%)**	91.8 ± 33.7	139.6 ± 32	*p* < 0.001
**vW Ag (%)**	105.4 ± 25	176.7 ± 51.8	*p* < 0.001
**D-dimers (ng/mL)**	318 ± 173	535.2 ± 552.9	*p* = 0.1
**Fibrinogen (mg/dL)**	284.6 ± 91.6	353.6 ± 165.5	*p* = 0.04

## Data Availability

Data supporting results can be found at the repository of 2nd Critical Care Department, Attikon University Hospital.

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
