# Peer review of "Platelet, Fibrinolytic and Other Coagulation Abnormalities in Newly-Diagnosed Patients with Chronic Thromboembolic Pulmonary Hypertension"

_diagnostics, 2022, doi:10.3390/diagnostics12051238_

Round 1

Reviewer 1 Report

The authors present their manuscript "Platelet, fibrinolytic and other coagulation abnormalities in newly-diagnosed patients with chronic thromboembolic pulmonary hypertension" for consideration of publication in Diagnostics. This study evaluate alternative diagnostic for chronic thromboembolic pulmonary hypertension by performing non-conventional hemostatic tests and markers of platelet activation and endothelial dysfunction.

This paper is well written and the experimental design is well indicated. However, I have few comments.

1- The global prevalence of CTEPH should be indicated in the study and the size of the sample must take account this prevalence. The sample s size is not enough to discuss the data.

2- Considering that almost all the CTEPH patients are from the Pulmonary embolic patient population, what are the differential diagnosis between CTEPH and PE patients? Do you think that your data can make a difference between  CTEPH and PE patients?

3- For reliable data it should be nice to have 3 groups in the study by adding the PE patients group.

4- Minor comments

-Typo in lines 41, 66, 204, 285, 387.

-line 402 " To the authors knowledge" replace it please by " To our knowledge"....

Author Response

We thank the reviewer for their insightful comments. Please find below our point-by-point response to the reviewer's comments regarding our manuscript “Platelet, fibrinolytic and other coagulation abnormalities in newly-diagnosed patients with chronic thromboembolic pulmonary hypertension”.

  • The global prevalence of CTEPH should be indicated in the study and the size of the sample must take account this prevalence. The sample s size is not enough to discuss the data.

Response: The manuscript has been revised in the following sections:

  • “CTEPH is a rare, chronic disorder, whose global incidence and prevalence rates have not been yet clearly assessed [4]. It has been estimated, nonetheless, that the annual incidence of CTEPH is approximately 4 cases / 100.000 population. [4]” (lines 45-47).
  • “One of this study’s main limitations is the small number of included patients. Nevertheless, our sample size is analogous with other similar studies in the field [18,20].”
  • Considering that almost all the CTEPH patients are from the Pulmonary embolic patient population, what are the differential diagnosis between CTEPH and PE patients? Do you think that your data can make a difference between  CTEPH and PE patients?

Response: The following revisions have been made:

  • “CTEPH is viewed as a long-term complication of pulmonary embolism (PE), affecting approximately 3% of PE survivors [5,6]. The disorder can be diagnosed when the clots in the pulmonary vasculature have not been resolved after more than three months of effective anticoagulant treatment [1]. CTEPH, however, can occur months or years after the acute PE, without the development of a new thromboembolic event in the meantime [4].” (line 47-52)
  • “Despite research advances, several aspects of the pathogenetic mechanisms of the disease remain elusive to date [9]. Initially it was thought that the unresolved thrombi were the cause of CTEPH [10]. It is now considered that endothelial dysfunction, vascular remodeling and vasoconstriction participate in the etiopathogenesis of the disease [3]. More specifically, evidence suggest that there is secondary microvasculopathy in the pulmonary vasculature of CTEPH patients, in the form of obstructive (but non-thrombotic) remodeling of the small pulmonary arteries, capillaries and venules [10]. These lesions appear to be similar to lesion founds in patients suffering from pulmonary arterial hypertension (e.g., plexiform lesions, intimal fibrosis and proliferation) [10].” (lines 59-67)

According to the current understanding of the pathophysiology of CTEPH, unresolved PE is considered as the initial event which triggers the development of the disorder. The etiopathogenesis of CTEPH, however, appears to be much more complex than this, and it has been suggested that it relies on secondary events as well (e.g., endothelial dysfunction, remodeling of the pulmonary vasculature, and coagulation abnormalities). Since different pathogenetic events could contribute to the genesis of PE and CTEPH (even though CTEPH is considered a complication of PE), and the naturally history of the two disorders portrays significant variations, the results of global coagulation assays between the two groups could be different. 

  • For reliable data it should be nice to have 3 groups in the study by adding the PE patients group.

Response: It is undisputed that such a comparison would provide valuable insights in the pathophysiology of the disorder. It could highlight the similarities and differences of the coagulation function in CTEPH and PE, thus potentially providing significant information on CTEPH’s pathogenetic processes. It is the intention of our research group to continue this research by comparing platelet and coagulation function in CTEPH and PE patients in a later time.

4- Minor comments

-Typo in lines 41, 66, 204, 285, 387.

Response: Corrected.

-line 402 " To the authors knowledge" replace it please by " To our knowledge"....

Response: Corrected.

Reviewer 2 Report

Comments to the Author

The manuscript titled with “Platelet, fibrinolytic and other coagulation abnormalities in 2 newly-diagnosed patients with chronic thromboembolic pul- 3 monary hypertension, trying to illuminate platelet function, fibrinolytic, thrombin generation 31 and other clot formation abnormalities of CTEPH patients. The topic itself is interesting and is suitable for this journal. In this study, however, there are several key remarks when it comes to the coverage and explanation.

The authors did not clearly explain:

  1. Endothelial dysfunction and vascular remodeling association with CTEPH
  2. Legend of Figure 1, 2 and 3.
  3. Sample collection method (2.2)
  4. The result part of the manuscript was not well written, and overall the presentation is very limited.

Minor comments:

  1. Add space in line number 41, 234, 285, 288, 298, 354, 384, 387, 395, 402.
  2. Author should add few graphs to represent the comparative result of different parameters of patient samples and healthy samples.

Author Response

We thank the reviewer for their insightful comments. Please find below our point-by-point response to the reviewer’s comments regarding our manuscript “Platelet, fibrinolytic and other coagulation abnormalities in newly-diagnosed patients with chronic thromboembolic pulmonary hypertension”.

The authors did not clearly explain:

  1. Endothelial dysfunction and vascular remodeling association with CTEPH

Response: The following revision was made:

“Despite research advances, several aspects of the pathogenetic mechanisms of the disease remain elusive to date [9]. Initially it was thought that the unresolved thrombi were the cause of CTEPH [10]. It is now considered that endothelial dysfunction, vascular remodeling and vasoconstriction participate in the etiopathogenesis of the disease [3]. More specifically, evidence suggest that there is secondary microvasculopathy in the pulmonary vasculature of CTEPH patients, in the form of obstructive (but non-thrombotic) remodeling of the small pulmonary arteries, capillaries and venules [10]. These lesions appear to be similar to lesions found in patients suffering from pulmonary arterial hypertension (e.g., plexiform lesions, intimal fibrosis and proliferation) [10].” (lines 59-67)

  1. Legend of Figure 1, 2 and 3.

Response: The following legends were revised

  • “Figure 1. Representation of an LTA aggregation trace.” (line 145)
  • “Figure 2. Representation of an LTA disaggregation trace” (line 161)
  • “Figure 3. Representation of a Rotem trace” (line 177)

  1. Sample collection method (2.2)

Response: The following section was revised:

“2.2        Sample collection method

Blood samples were obtained from the patients’ pulmonary artery during a right-heart catheterization performed for the diagnosis of CTEPH. Blood samples from the healthy participants were collected from an antecubital vein.” (line 101-104).

  1. The result part of the manuscript was not well written, and overall the presentation is very limited.
  2. Minor comments: Author should add few graphs to represent the comparative result of different parameters of patient samples and healthy samples.

Response to points 4 and 5: The “result” section has been rewritten and each assay’s results were presented in separate paragraphs. 12 graphs that represent the comparative results of all the parameters that presented statistically significant results between the two groups have also been added (lines 319-410).

  1. Minor comments: Add space in line number 41, 234, 285, 288, 298, 354, 384, 387, 395, 402.

Response: Corrected.

Reviewer 3 Report

Authors analyzed  40 newly-diagnosed CTEPH patients and 35 healthy controls: they should explain why this cohort of patients it is significant. Authors should compare more studies also found in literature justifuing results. I suggest to summarise results in a final table (discussion section).

In order to improve the state of art, authors should add more references about perspectives such as the use of further approach for data analysis such as (artifcial intelligence):

doi:10.3390/jcm8030360 

https://doi.org/10.3390/healthcare10020343

doi: 10.23919/AEIT.2018.8577362. 

and other similar ones (to add in the introduction section) 

Authors should add a conclusion section. 

Important: authors should enhance the difference between the proposed paper and the following one:

https://www.tandfonline.com/doi/full/10.1080/09537104.2018.1499890

Author Response

We thank the reviewer for their insightful comments. Please find below our point-by-point response to the reviewer’s comments regarding our manuscript “Platelet, fibrinolytic and other coagulation abnormalities in newly-diagnosed patients with chronic thromboembolic pulmonary hypertension”.

  1. Important: authors should enhance the difference between the proposed paper and the following one: https://www.tandfonline.com/doi/full/10.1080/09537104.2018.1499890

Response: The following revisions were made:

  • “In a previous study conducted by our group [12], we assessed platelet and coagulation processes in newly-diagnosed patients with pulmonary arterial hypertension using non-conventional coagulation assays, that portrayed platelet dysfunction and thrombin generation deficits in the patient population. Since there could be common mechanisms that contribute in the development of both pulmonary arterial hypertension and CTEPH that could affect hemostasis (e.g., endothelial dysfunction and vascular remodeling) [10], in this study we assessed platelet and coagulation processes in the CTEPH population.” (line 71-77)
  • “In our previous study where we studied platelet and coagulation processes in pulmonary arterial hypertension [12], we showed that the patient population also presented diminished platelet aggregation, prolonged initiation of the clotting process and thrombus formation, and diminished total amounts of free thrombin and maximum thrombin concentrations. Since the pathogenetic processes of both pulmonary arterial hypertension and CTEPH haven’t been clarified to date and the role that coagulation deficits play in the development and progression of pulmonary hypertension haven’t been elucidated [4,10], it could be difficult to speculate if those findings are characteristics of pulmonary hypertension or an epiphenomenon [30].“ (lines 488-496).
  1. Authors analyzed 40 newly-diagnosed CTEPH patients and 35 healthy controls: they should explain why this cohort of patients it is significant.

Response: The following section has been added

“Despite research advances, several aspects of the pathogenetic mechanisms of the disease remain elusive to date [9]. Initially it was thought that the unresolved thrombi were the cause of CTEPH [10]. It is now considered that endothelial dysfunction, vas-cular remodeling and vasoconstriction participate in the etiopathogenesis of the dis-ease [3]. More specifically, evidence suggest that there is secondary microvasculopathy in the pulmonary vasculature of CTEPH patients, in the form of obstructive (but non-thrombotic) remodeling of the small pulmonary arteries, capillaries and venules [10]. These lesions appear to be similar to lesions found in patients suffering from pulmonary arterial hypertension (e.g., plexiform lesions, intimal fibrosis and prolifera-tion) [10]. Literature data also support that abnormal platelet function and ineffective fibrinolysis may play a key role in the development of the disorder [9,11]. Nevertheless, original research data portraying platelet and coagulation abnormalities related to CTEPH remain limited.” (lines 59-71)”

  1. Authors should compare more studies also found in literature justifuing results.

Response: The following references were added: 4,10,30

  1. I suggest to summarise results in a final table (discussion section).

Response: The results of the study were summarized in the discussion section:

“More specifically, platelet aggregation was shown to be diminished, a significant pro-portion of the patients presented decreased platelet aggregate stability, the initiation of the clotting process and clot propagation were both prolonged, the rate of fibrinolysis was diminished, and thrombin generation capacity was defective. Regarding platelet activation and endothelial dysfunction markers, TXA2, p-selectin, VWF:Ac and VWF:Ag were found to be increased.” (lines 414-419)

  1. Authors should add a conclusion section.

Response: A conclusion section was added, as follows:

“5. Conclusion

The results of this study suggest that CTEPH patients present platelet function, fibrinolytic, thrombin generation and other clot formation abnormalities. Even though the research on the field of the pathobiological background of the disorder is rapidly evolving, the complex effects that coagulation may have in the progression and development of the disorder haven’t been yet fully understood. Well-designed clinical studies are needed to further evaluate hemostatic abnormalities in the CTEPH setting and assess their potential clinical applications.” (lines 504-511)

  1. In order to improve the state of art, authors should add more references about perspectives such as the use of further approach for data analysis such as (artifcial intelligence): doi:10.3390/jcm8030360 , https://doi.org/10.3390/healthcare10020343, doi: 10.23919/AEIT.2018.8577362, and other similar ones (to add in the introduction section)

Response: We would like to thank you for these very interesting suggestions. Nevertheless, the specialties of our research group fall under the categories of pulmonology and laboratory hematology. We feel, therefore, that we are not sufficiently equipped to truly evaluate all these cutting-edge techniques, or the way they could impact the field of laboratory hematology in the future.

Round 2

Reviewer 1 Report

 All my comments have been solved. I do not have over comment.

Author Response

We would like to thank the reviewer for their kind consideration of our work. 

Reviewer 2 Report

In the article “Platelet, fibrinolytic and other coagulation abnormalities in newly-diagnosed patients with chronic thromboembolic pulmonary hypertension” authors considered reviewer comments nicely. Few points are still unclear.

Comments for authors:

  1. Author contribute more about the sample collection like
    1. Vial type
    2. Collection method
    3. Sample storage conditions
  2. Authors not contribute about the legend of the figure 1, 2, and 3.
  3. Author add the reference of each section of the table 1 in the separate column.
  4. In line 204 space correction
  5. In section 2.3.6 authors can add about the different markers and their receptor or mention their references. Why authors select these markers?

Author Response

We would like to thank the reviewer for their kind consideration of our work. Please see bellow our point-by-point response to the reviewer comments:

  • Author contribute more about the sample collection like
  1. Vial type

Response: The following revision was made:

“The blood specimens were placed in blood collection tubes containing 3.8% trisodium citrate.” (lines 105-106).

  1. Collection method

Response: The text was revised as follows:

“Blood samples were obtained from the patients’ pulmonary artery during a right-heart catheterization performed for the diagnosis of CTEPH. More specifically, the collection was made from the distal lumen of the pulmonary catheter with very slow aspiration. Blood samples from the healthy participants were obtained from the antecubital vein.” (lines 104-106).

  1. Sample storage conditions

Response: The revised text is located in lines 112-118:

“PFA-100, LTA and ROTEM assays were performed within three hours from the collection of the specimens. The ETP assay and the tests assessing platelet activation and endothelial dysfunction laboratory markers were performed from plasma obtained after the whole blood samples were centrifuged. The removed plasma was snap frozen and stored in laboratory fridges at the temperature of −20 °C (with round-the-clock monitoring of the freezer temperatures). These tests were performed at a later time.”

  • Authors not contribute about the legend of the figure 1, 2, and 3.

Response: The results of the ROTEM and LTA assays are provided by the respective analyzers not only as numerical outcomes, but also as graphical representations called “traces”.

In Figures 1 and 3 we provide a graphical illustration of what a normal trace produced by these analyzers would look like.

For example, regarding LTA in lines 132-133 we write that “The results of the LTA testing are presented as a trace and the measured parameter is maximal aggregation (%).”

We revised the legend of Figure 1 as follows: “Graphical illustration of a normal LTA aggregation trace.” (line 148).

In Figure 2 we present a graphical illustration of an LTA trace that shows disaggregation.

The legend of figure 2 is revised to: “Graphical illustration of an LTA trace showing disaggregation.” (line 164).

Regarding ROTEM we write in line 171: “The results of this assay are depicted as a trace”.

The legend of Figure 3 is revised as follows: “Graphical illustration of a normal Rotem trace” (line 180).

  • Author add the reference of each section of the table 1 in the separate column.

Response: The interpretation of the ROTEM parameters is provided by algorithms, that explain which factors affect which parameter. We added references 17 and 18 that thoroughly analyze these algorithms.  The revised text is located in line 185: “Table 1. Rotem’s major parameters and their interpretation [17, 18].”

  • In line 204 space correction

Response: Corrected

  • In section 2.3.6 authors can add about the different markers and their receptor or mention their references. Why authors select these markers?

Response: The following revisions were made:

  • “TXA2 is produced by activated platelets and is a well-known marker of platelet activation [20].” (lines 213-214)
  • “p-selectin functions as an adhesion molecule, and is stored in the a granules of platelets and in the Weibel–Palade bodies of endothelial cell [21]. It is frequently used in research as a marker of both platelet activation and endothelial dysfunction [21].” (lines 216-218)
  • “Serotonin is produced by activated platelets [22]. In the coagulation setting, serotonin promotes vasoconstriction and platelet aggregation [22].” (lines 221-222)
  • “von Willebrand factor, a glycoprotein synthesized by endothelial cells, facilitates platelet adhesion to the injured vascular subendothelium [23]. It is widely used as a marker of endothelial dysfunction in vascular disorders [23].” (lines 225-227)

Reviewer 3 Report

The paper is now ready for publication

Author Response

We would like to thank the reviewer for their kind consideration of our work